# An IoT Platform towards the Enhancement of Poultry Production Chains

**DOI:** 10.3390/s20061549

**Published:** 2020-03-11

**Authors:** Iker Esnaola-Gonzalez, Meritxell Gómez-Omella, Susana Ferreiro, Izaskun Fernandez, Ignacio Lázaro, Elena García

**Affiliations:** 1TEKNIKER, Basque Research and Technology Alliance (BRTA), C/ Iñaki Goenaga 5, 20600 Eibar, Spain; meritxell.gomez@tekniker.es (M.G.-O.); susana.ferreiro@tekniker.es (S.F.); izaskun.fernandez@tekniker.es (I.F.); ignacio.lazaro@tekniker.es (I.L.); elena.garcia@tekniker.es (E.G.); 2Faculty of Informatics, University of the Basque Country (UPV/EHU), Paseo Manuel Lardizabal 1, 20018 Donostia-San Sebastián, Spain

**Keywords:** IoT, agriculture, poultry welfare

## Abstract

As a consequence of the projected world population growth, world meat consumption is expected to grow. Therefore, meat production needs to be improved, although it cannot be done at any cost. Maintaining the health and welfare status of animals at optimal levels has traditionally been a main concern of farmers, and more recently, consumers. In this article the Poultry Chain Management (PCM) platform is presented. It aims at collecting data across the different phases of the poultry production chain. The collection of this data not only contributes to determine the quality of each phase and the poultry production chain as a whole, but more importantly, to identify critical issues causing process inefficiencies and to support decision-making towards the holistic improvement of the production chain. Results showed that the information gathered can be exploited to make different suggestions to guarantee poultry welfare, and ultimately, improve the quality of the meat.

## 1. Introduction

The population of the world is growing at exponential rates and, according to United Nation’s Revision of World Population Prospects https://esa.un.org/unpd/wpp/), it is projected to reach a number of over 9.7 billion by 2050. This population growth poses issues that may affect the sustainability of demographic, social and economic systems. More specifically, one of the main challenges consists of finding a way to feed all these people and agriculture, which can be understood as the cultivation and breeding of animals and plants to provide food and other products to sustain and enhance life, plays a vital role in addressing this issue.

Due to the aforementioned population growth, world food consumption will increase accordingly, and consequently, so will meat consumption. As a matter of fact, meat consumption per capita is expected to increase from 38.7 kg in 2005, to 49.4 kg in 2050 and overall meat consumption, is expected to grow by 70% in the period of 2000–2030 and by 120% in the period of 2030–2050 [1]. Meat production is one of the most important sectors at a worldwide agriculture level and also in Europe. According to Eurostat http://ec.europa.eu/eurostat/statistics-explained/index.php/Meat_production_statistics there are almost 7 million livestock farms in the EU, and the four main types of farms are those rearing pigs, bovine animals, poultry, and sheep and goats. However, in order to satisfy the future meat demand increase, there is a dire need to increment meat production.

However, this meat production improvement cannot be made at any cost. Maintaining the health and welfare status of livestock at optimal levels has traditionally been a main concern of farmers [2], and more recently, consumers [3,4,5]. Comfort is one of the main factors that influences the well-being and health of animals during their lifetime [6], hence it cannot be neglected. Providing a comfortable environment not only maximizes the profit garnered from each animal, but also reduces mortality, which in turn allows the amount of wasted water and feed resources to be reduced. However, ensuring livestock comfort within farms may not be enough to guarantee their well-being and optimal meat quality when arriving at slaughterhouses. The way animals are transported from farms to slaughterhouses and the way they are loaded and unloaded in transport vehicles are stressful operations that might affect welfare and increase animal mortality [7]. Therefore, all these facts reinforce the need to ensure comfort of livestock through the whole production chain, from their breeding in farms until their arrival at slaughterhouses.

In the context of the Internet of Food & Farm 2020 (IoF2020) H2020 project https://www.iof2020.eu/, one of the trials is aimed at optimizing animal health, production chain transparency and traceability. Within this trial, there is a poultry production chain use case. In 2014, poultry meat production in the EU reached 10.5 million tonnes, representing around 12% of world production [8] and, similar to the meat production of other livestock farms, the poultry meat production is expected to grow at a worldwide level, reaching 181 million tonnes in 2050 [1]. 70% of EU poultry meat production is concentrated in seven member states (Poland, France, United Kingdom, German, Spain, Italy and the Netherlands) and chicken consumption has overtaken that of the pork or beef in some places such as the USA https://www.nationalchickencouncil.org/about-the-industry/statistics/per-capita-consumption-of-poultry-and-livestock-1965-to-estimated-2012-in-pounds/.

Nowadays, poultry chain managers get paid according to the quality of the meat that is obtained at the slaughterhouse. Therefore, one of the main criteria used to evaluate the whole production chain is the final quality of the meat. However, the quality of meat is strongly influenced by the stressful situations to which poultry is exposed throughout the production chain phases [9], even though market pressure does not sufficiently encourage breeding companies to give welfare traits greater weighting in their programs [8]. Consequently, a platform where information throughout the whole production chain is collected, establishes how to exploit that information to evaluate the quality of each phase and support the decision-making towards improving the final meat quality while ensuring animal welfare. Nevertheless, the development of such a platform is not an easy task due to the different inherent characteristics of the data sources (e.g., heterogeneity in terms of format and structures) to be integrated, and the need to ensure a secure data exchange environment.

In this article the Poultry Chain Management (PCM) platform is presented. It aims at collecting data across the different phases of the poultry production chain. The collection of this data not only contributes to determine the quality of each phase and the poultry production chain as a whole, but more importantly, to identify critical issues causing process inefficiencies and to support decision-making towards the holistic improvement of the production chain. Although it is motivated by the IoF2020 project, the PCM platform is based on open-source components and open standards towards its replicability in other projects and other livestock production chains. The rest of this article is structured as follows. Section 2 reviews related work. Section 3 presents the PCM platform and describes the flow of the data throughout the whole poultry production chain. Section 4 explains the KPIs used to evaluate each poultry production chain phase and showcases them in a real-world use case. Section 5 demonstrates the exploitation of the collected data with an analytical tool. Finally, the conclusions of this work are presented in Section 6.

## 2. Related Work

Nowadays, food industries invest a considerable part of their resources to ensure the quality of their products [10], and the poultry industry is no exception. As a result, many research efforts have been dedicated to studying the influence that situations occurring in the different poultry production chain phases have in the poultry welfare and final meat quality.

Different solutions are proposed for ensuring poultry welfare and good rearing conditions within farms. A study used infrared images to discover that poultry under cold stress conditions spent about four times more energy trying to maintain body temperature [11]. Infrared thermography has also been used to evaluate metabolic heat loss of poultry fed with different energy densities [12]. There is work that develops predictive models to predict poultry growth [13] and death rates [14] within farms. Furthermore, the effect the indoor conditions of the farm has on poultry welfare has been researched. A laboratory experiment determined the influence of different temperature and humidity combinations on the physiological response of poultry [15]. Another study proposed the diagram shown in Figure 1, indicating how the relationship between ambient temperature and relative humidity affects poultry heat stress [16]. Towards ensuring optimal poultry rearing conditions, there is a system combining Knowledge Discovery and Semantic Technologies [17]. This system sends farmers notifications when a stressful thermal situation is predicted, so that they can anticipate such undesirable situations by taking actions on the farm beforehand.

Few scientific studies have been conducted to analyze the effect that catching and loading has on the welfare of poultry, although it is reasonable to assume that it generates a particular pattern and level of injury and stress. A study recorded reduced heart rates and catching damage in birds when using a semi-automatic loading system, in comparison to 3 other manual catch and loading approaches [18]. Other studies compared the death rates between a mechanical poultry catching system and a manual one, registering an increase in the manual system [19,20]. Furthermore, the rate of injuries has also been compared between manual and automatic poultry catching systems. There are studies that recorded significantly reduced injuries for mechanically caught poultry (3.1%) compared with manually caught (4.4%), especially with respect to leg injuries [19]. However, another study found no differences in the percentage of bruises, meat quality or corticosterone levels between manual and automatic catching methods [20]. There are studies that also attribute the decrease in injury rates to the reduction in speed of conveyors in automatic loading systems [19]. According to another study, the slower catching of automatic systems compared with manual catching may result in more heat stress mortality [21]. Furthermore, a study showed no significant difference between catching poultry from one leg (grabbing 2–3 birds per hand) or two legs (grabbing a maximum of 2 birds per hand) [22]. Additionally, the benefits of training and incentivising farm operators to adequately interact with animals in farms are recognized [23,24,25]. From these studies it can be concluded that, when properly carried out, using optimal equipment and trained personnel, both methods can result in low levels of injury and low levels of stress to the birds. Conversely, both manual and mechanical catching can result in unacceptably high levels of bruises, fractures and other injuries, as well as high stress levels, if carried out in an improper way [26].

Although loading procedures are more likely to cause physical injuries, the transport phase has also been reported to be stressful to poultry and have a direct effect on final meat quality. A research focused on the effect of truck temperature and relative humidity [27]. A study conducted with simulated electronic chickens suggested that ambient temperatures in the range of 11 °C to 25.1 °C during transport phases are thermally comfortable as they allow poultry to regulate heat with their metabolism in order to stay comfortable [28]. Conducted research also shows that poultry deaths on transport differs depending on the geographical point and the season of the year. For example, a study reported that poultry deaths increased from 0.28% in winter, to 0.42% in summer for transport in a subtropical climate of southeastern Brazil [7]. Another research, in this case conducted in Italy, showed that death rates in transport increased from 0.35% in winter months to 0.47% in summertime [29]. Transport duration has also a direct effect on the poultry welfare [30], as a study showed that the lowest poultry death rates happened in journeys of less than 2 h (0.29%) and the highest when transport times exceeded 5 h (0.46%) [31]. Additionally, in another different research, death rates also increased from 0.24% (in routes of less than three and a half hours) to 0.45% (in routes of more than 5 h) [32]. Likewise, the transport distance traveled from the farm to the slaughterhouse influences the welfare of the poultry. A study conducted in the Czech Republic recorded that for journeys up to 50 km death rate was 0.15%, and 0.86% for distances exceeding 300 km [33]. Last but not least, the vibrations related with the sudden acceleration of trucks have also been studied and showed a negative effect on poultry welfare due to the impacts and muscle strain suffered [34,35,36].

A certain number of individuals from each flock may be rejected in the slaughterhouse due to status as a result of poor welfare conditions, such as abnormal levels of contact dermatitis, parasitism or systemic illness in the house [21]. Other causes for rejection include infectious diseases such as colisepticaemia [37], skin injuries such as cellulite or dermal squamous cell carcinoma [38], and fractures or bruises [39]. Furthermore, the tasks performed in the slaughterhouse phase may also result in the rejection of the animal, such as contamination produced by evisceration [39].

According to the European Food Safety Authority, a monitoring system for animal welfare is made up of the following steps: (I) identification of the goal; (II) identification of the population concerned, and definition and selection of the survey population; and (III) selection of the indicators and the systematic collection of data [40]. Furthermore, the systematic collection of animal-based measures and its subsequent storage in well-defined databases could pave the way to better assessing the validity and robustness of those measures, thus moving towards quantitative risk assessment of animal welfare [41]. Moreover, in animal health research, the visualization of relationships between risk factor and health outcomes is based on the correction of indicators, followed by an analysis to investigate relevant associations. There are data collection systems at specific poultry production chain phases, especially in the breeding phase [42,43,44,45]. However, to the extent of our knowledge, there is no platform that collects information throughout the whole poultry production chain for its exploitation towards poultry welfare improvement.

## 3. The Poultry Chain Management Platform

Chickens raised for meat (also known as broiler chickens) arrive at the farm around 21 days of after hatching and it typically takes around 7 weeks until animals reach the proper size and weight to be sent to the slaughterhouse https://www.chickencheck.in. In order to ensure adequate growth and welfare, there are dietary needs and comfort requirements that need to be fulfilled. On the one hand, these comfort requirements vary with age. For example, the first days of breeding, poultry require a higher ambient temperature because their body temperature, metabolic rate, insulation from feathering and thermoregulatory ability are relatively low [46,47]. On the other hand, these comfort requirements may also change if, for certain reasons, the growth of the poultry flock is slower or faster than expected. This phase, where poultry are raised, is referred to as the breeding phase. Once the flock reaches the determined size and weight and they are ready to be sent to the slaughterhouse, farm operators catch and load them into holding cages or modular bins. These cages are specifically designed to ensure that birds cannot hurt themselves or other birds, and that the air is able to circulate. This phase is referred to as the loading phase. These cages are then placed on a truck which transports the flock from farms to the slaughterhouse. This is the transport phase. Finally, once the animals arrive at the slaughterhouse, they are unloaded and handled by slaughterhouse operators in the phase referred to as the slaughterhouse phase.

Summarizing, there are four phases that comprise the whole poultry production chain from when baby chicks arrive at the farm until they arrive at the slaughterhouse: breeding, loading, transport and slaughterhouse phases. Figure 2 depicts the four main poultry production chain phases.

The variety of situations that may occur throughout the different phases has a direct impact on the health and welfare of animals, as well as on the quality of their meat by the time they arrive at the slaughterhouse. Therefore, monitoring each of these phases can provide relevant data that can later be exploited to detect causalities behind these issues and adopt measures towards the holistic improvement of the poultry production chain. Therefore, a platform that collects and manages information across the whole production chain is considered of utmost importance.

### 3.1. The PCM Platform Workflow

The PCM is a cloud-based platform supported by FIWARE open-source components, and implemented based on open standards in charge of collecting, processing and storing data coming from different phases of the poultry production chain in a secure way. The PCM platform is divided into four different stages as can be seen in Figure 3. It is worth mentioning that the exploitation of the collected information is left out of the architecture.

Data Sources

This is the first stage of the platform where methods to access, assess, convert and aggregate signals are employed by different devices, machines or systems to represent real-world parameters as communicable data assets. In the context of the poultry production chain, data sources are heterogeneous ranging from sensors to measure the environmental conditions within farms or trucks, to wristbands measuring the way operators handle animals. Furthermore, each data source may have its own method to collect the information.

Data Acquisition

This is the stage where the data coming from the various monitoring devices and systems is received. This stage consists of different agents and components to enable the adequate data handling coming from heterogeneous data sources.

The central component is the FIWARE Orion Context Broker https://fiware-orion.readthedocs.io/ (OCB), a C++ implementation of the NGSIv2 REST API binding. The OCB allows the management of the entire lifecycle of context information including updates, queries, registrations and subscriptions.

The recurrent news about security breaches, private data leaks and the inappropriate use of data, makes the security of the IoT platforms a vital requisite nowadays. Therefore, the security of the PCM platform is of utmost importance. Due to the multitude of different devices, sensors and services involved in the data flow, the security of the PCM platform has to be handled by different agents. FIWARE Keyrock https://fiware-idm.readthedocs.io/ is responsible for identifying, authenticating and authorizing devices and systems to publish their information in the OCB, by associating them rights and restrictions with established identities. It is based on OpenStack Keystone https://docs.openstack.org/keystone, a service that provides API client authentication, service discovery, and distributed multi-tenant authorization by implementing OpenStack’s Identity API, and OpenStack Horizon https://docs.openstack.org/horizon/, which provides a web-based user interface to OpenStack Keystone. Additionally, to complete the security module of the PCM platform, the FIWARE PEP Proxy https://fiware-pep-proxy.readthedocs.io/ provides a security layer for adding authentication and authorization filters, and it is combined with Keyrock to enforce access control to backend applications.

Furthermore, data coming from IoF environmental sensors (explained in Section 3.2) is collected by a FIWARE IoT Agent https://fiware-iotagent-json.readthedocs.io/, which has a Mosquitto MQTT broker https://mosquitto.org/ embedded. The MQTT protocol is envisioned as a high performing solution for data acquisition, not only because of the low power and memory needed for the implementation of different clients in small devices, but also due to the low bandwidth needed. The FIWARE IoT Agent is used as a bridge to publish sensor data to the central OCB.

Last but not least, there is an Apache Tomcat server in charge of executing periodic tasks for sending CSV-like files to store them in file structures, and get data from external sources (e.g., meteorological prediction from weather forecasting services) in order to publish it in the central OCB.

Data Processing

This is the stage where the data acquired in the previous stage is sent to be stored in the corresponding data repository. The main component of this stage is a FIWARE Cygnus https://fiware-cygnus.readthedocs.io/ agent. Cygnus is a connector in charge of storing certain sources of data in certain configured third-party storage, creating a historical view of such data. Internally, Cygnus is based on Apache Flume http://flume.apache.org/, a technology addressing the design and execution of data collection and storage agents. An agent is composed of a listener or source in charge of receiving the data, a channel where the source puts the data once it has been transformed into a Flume event, and a sink, which takes Flume events from the channel in order to store the data within its body into a third-party storage.

Data Storage

This is the stage where the collected data is stored and remains accessible for its future exploitation. Two main data storage repositories are considered in this stage. On the one hand, MongoDB https://www.mongodb.com/, a NoSQL database which uses JSON-like documents and is the adequate option to store data coming from heterogeneous sources. On the other hand, a file structure to store files created in the different poultry production chain phases and that are not considered to be worth passing through the OCB for various reasons. For example, the amount of data acquired during the loading of the animals with the wearable devices is too big, and in this case, the CSV structure is more suitable for data analysis tasks.

### 3.2. the Information Collection from the Different Phases

The poultry production chain is characterized by four phases: breeding, loading, transport and slaughterhouse phases. In each of these phases, different information is retrieved, which could potentially be exploited to both evaluate the quality of each phase and support the decision-making towards improving the final meat quality while ensuring animal welfare. This section explains the information collected from each phase, and the way in which data from each phase is obtained and stored in the PCM platform.

Breeding Phase

This is the phase where chickens arrive at the farm 21 days after hatching and spend around 7 weeks until they reach the required weight and size. During these 7 weeks the comfort requirements of the flock including temperature and humidity vary, and if they are not satisfied, animals may be exposed to stressful situations that may result in deficient growth. For example, if relative humidity is too low, there is a higher production of dust and an increase in the number of airborne microorganisms, which may increase susceptibility to respiratory diseases especially during the early days of the broiler. However, if relative humidity levels which are too high are combined with high temperatures, broilers may die from hyperthermia or hypoxia [26]. Although there are different guidelines to set minimum comfort requirements, in the context of the IoF2020 project, they are defined by farmers, and they can be modified at any moment during the breeding phase for different reasons, including a slower or faster rate of flock growth.

The information generated in the breeding phase is retrieved from different data sources. On the one hand, IoF environmental sensors are installed throughout the farm to measure the climatic conditions in different points of the farm. In the initial deployment plan of IoF2020 project, Tibucon sensors (which were developed as part of the TIBUCON FP-7 project https://cordis.europa.eu/project/rcn/95501/en) were deployed, which measured temperature, humidity and luminosity values. These sensors were then upgraded in terms of hardware and software, and CO_2_ and ammonia level measurement capabilities were added. The resulting IoF environmental sensors are easy-to-install, have a battery supply and offer wireless connectivity. They measure farm conditions every 30 s and this data is sent through a low-power multi-hop wireless network based on the standard IEEE 802.15.4 https://standards.ieee.org/standard/802_15_4-2015-Cor1-2018.html) to a gateway installed in the control room next to the farm. The gateway then sends the data to the Mosquitto MQTT Broker of the PCM platform which will in turn send it to the central OCB.

In addition, there is an existing third-party Farm Management System deployed within farms. This system collects information from different devices and sensors installed within farms, including weight scales, temperature and humidity sensors. The information collected by these sensors is stored in a centralized database. In order to integrate this information within the PCM platform, a Visual Basic application periodically retrieves the latest data stored in the central database of the Farm Management System and sends it to the FIWARE PEP Proxy. Once this Proxy authenticates and authorizes the data delivery, it is forwarded to the central OCB.

The external weather conditions have a direct impact on the indoor conditions of the farm. Therefore, a weather forecasting service is leveraged to retrieve this information. More precisely, Tiempo.com https://www.tiempo.com/ service’s API is accessed executing a Java-based periodic task which collects the weather forecasting for the location where the IoF2020 project’s use case farms are located. Tiempo.com offers predictions with different time-horizons and formats, among which XML files with hourly predictions for the next 5 days are leveraged. These files include the predicted temperature, relative humidity, sky status (e.g., cloudy or sunny) and wind speed for a given location, and this data is sent to the central OCB.

Loading Phase

In most European countries, poultry loading is performed by catching the birds by one or two legs and carrying three or four birds in each hand [48,49]. The poultry handling is recommended to be carried out in a careful and conscientious way in order to avoid stressful situations, injuries and subsequent downgrading of the meat. However, in practice, the loading phase is often rather rough due to the poor working conditions of the personnel, consisting of arduous and repetitive work in a dusty environment [50]. In summary, the loading of animals into transport trucks has a direct effect on the final quality of the meat and its supervision may ensure poultry welfare and prevent injuries.

In order to monitor how the flock is loaded into trucks, operators wear electronic wristbands which measure arm sway acceleration. It is worth mentioning that operators work in an environment without connectivity, so wristbands cannot send data periodically and are required to have large data storage capabilities. Initially, operators used Wear OS smart watches https://wearos.google.com/, however, they were later replaced by Axivity AX3 3-axis logging accelerometer wristbands https://axivity.com/product/wrist-band due to difficulties capturing data at a continuous rate. The wristbands collect information during the whole loading phase (which may last from 3 to 5 h) , and then these wristbands are sent to the slaughterhouse in the transport trucks. Once they arrive at the slaughterhouse, a farm operator plugs the wristbands into the USB port of a PC and the collected information is managed using the OMGUI (Open Movement GUI) software. This software allows both the visualization and the download of the collected data into a CWA binary format file. This file is not compatible with Microsoft Excel or other third-party software, so the collected data needs to be exported into a CSV file using the OMGUI. As these CSV files may be too large (up to 4 GBs of data), the storage of information through the OCB is discarded. Instead, these CSV files are sent via WeTransfer to the PCM platform managers. Next, they store these files in the adequate folder of a file structure where they remain accessible to be exploited.

Transport Phase

This is the phase where the poultry flock is transported from farms to slaughterhouses. The transport phase has become a cause for concern because of animal welfare consideration, associated chicken mortality and consequential economic losses [9,51]. The transport journey duration is directed linked to the fasting duration of the poultry , so it has to be correctly estimated to ensure that the feed privation will be as short as possible. As a matter of fact, domestic birds can be transported without food and water up to 12 h [52]. Furthermore, the transport vehicle must ensure the safety of the animals and their welfare [53], for example by using side covers to protect birds from cold and wet weather while not impeding the air circulation. Last but not least, the driving style is directly related to the amount of stress perceived by poultry. Smooth and consistent speed driving habits allow the poultry to relax more during a journey, thus ensuring their welfare and the final meat quality [54]. Therefore, a good transport preparation is essential to avoid causing different degrees of injuries and stress to poultry, ranging from mild discomfort to more severe situations that may terminate in death.

Trucks transporting animals from farms to slaughterhouses are equipped with sensors that measure certain environmental properties. These are IoF transport sensors (similar to the IoF environmental sensors installed within farms) and they are attached to the cages or modular bins where poultry is previously loaded. The IoF transport sensors measure temperature, humidity, acceleration, ammonia and CO_2_ values. Taking into account that the transport trucks are vehicles without connectivity, sensors cannot send the collected information to the MQTT broker in real-time, but instead, they record all the information until the trucks arrive at the slaughterhouse. Once there, sensors are unloaded from the truck and placed near the slaughterhouse gateway. When the sensor detects the wireless network created by the gateway, the previously recorded information is sent to it, and afterwards, to the MQTT broker. From there, to the OCB and then, data is stored in MongoDB.

Slaughterhouse Phase

This is the phase where, once the poultry flock is unloaded from transport trucks, they are processed and packaged. This stage includes, in turn, different sub stages [55]. First of all, chickens are slaughtered and completely bleed. Then, in the scalding step, carcasses are immersed into hot water to ease the elimination of feathers. Afterwards, the carcasses are submitted to gutting and washing processes, and carcasses are classified according to their weights and quality. This process is followed by the chilling of carcasses and entrails. Finally, the carcasses are sent to the cutting stage where different pieces or products of poultry meat (e.g., wings and breasts) are produced and packaged. Therefore, the overall quality of the slaughterhouse phase is based on the quality of the final poultry meat.

Once the transported poultry flock is unloaded in the slaughterhouse, operators evaluate their state. They take a random sample of 200 chickens of which the number of dead animals and physical conditions (e.g., broken wings and bruises) are assessed. When the flock is processed and packaged, the status of the plucking or evisceration is also evaluated. The overall quality of the slaughterhouse phase is assessed based on the aforementioned criteria and it is registered in an Excel file. This file contains information of flocks coming from different farms and belonging to different poultry production chains and it is sent to the PCM platform managers once a week. Afterwards, this file is stored in the file structure of the platform. It is worth mentioning that, within this file, each flock is correctly identified so that it can be related to the corresponding information of the previous phases.

## 4. The Poultry Chain Quality Indicators

The data collected throughout the different phases of the poultry production chain remains accessible in the storage systems of the PCM platform. As a matter of fact, this data is retrieved and exploited by different data analytic services for various purposes. One of those services generates quality indicators for each phase. These indicators are used to determine the overall quality of the poultry production chain, and may support the adoption of specific actions towards the improvement of future production chains.

An indicator is an objectively verifiable measurement which reflects the activity, assumption, or effect being measured and allows for comparisons both between different populations or individuals and between measures of the same population or individual at different points in time [56]. Furthermore, the crucial factors when defining an indicator are that it is valid (or appropriate) and reliable (or trustworthy) as well as feasible to measure, given the resource constraints [40]. This section details the exploitation of the data available in each phase for developing the indicators used to determine their quality. In addition, a real-world poultry production chain use case is employed to showcase the extraction and meaning of such indicators. Namely, one that started on 30 April 2019 and ended on 01 July 2019 with a flock of around 33,000 chickens raised in the poultry farm shown in Figure 4. For each of the four phases, the conditions evaluated are described, defined Key Performance Indicators (KPIs) are specified and the results obtained for the aforementioned use case are detailed.

The KPI generation for each phase is automatized with R scripts that are automatically executed in an Rserve https://www.rforge.net/Rserve/ version 3.2.5 deployed in a Docker https://www.docker.com/ container. The only exception are the slaughterhouse phase KPIs, which are manually calculated by slaughterhouse operators and saved in an Excel file. Furthermore, obtained KPIs are stored in the MongoDB database of the PCM platform, thus remaining accessible for their analysis or further exploitation. An example of the generated exploitation of the KPIs is described in Section 5.

### 4.1. Poultry Breeding Phase

In order to determine the quality of the poultry breeding phase, five KPIs are defined. The first set of KPIs comprises two KPIs which are calculated based on the optimal farm temperature for the flock rearing. These temperatures, which are provided by farmers, vary for the different poultry growth stages (e.g., chicklings or adult stages) and other factors (e.g., a slower or faster growth pace of the flock). Namely, the aforementioned two KPIs are *Temperature Warning* and *Temperature Alarms*. The *Temperature Warnings* determines the percentage of time when farm temperatures deviate between 1.5 °C and 3 °C from the optimal temperature. This KPI determines the period of time in which animals have been exposed to mild uncomfortable temperatures while they are in the farm. *Temperature Alarms* is the percentage of time when farm temperatures deviate more than 3 °C from the optimal temperature. When this situation occurs, it is considered that the flock is exposed to severe thermal discomfort that may terminate in considerable heat stress. Therefore, this KPI determines the percentage of time in which animals have been exposed to severe uncomfortable temperatures in the farm. The second set of KPIs is based on the HIS, which combine the effects of both temperature and relative humidity of the air to determine the stress to which the flock is exposed within the farm [16]. Depending on the value of the HIS, a level of stress is assumed for the flock. A HIS value between 70 and 75, establishes the *Alert Situation* KPI where poultry may start to pant. A HIS value between 76 and 81 is considered a dangerous situation (*Danger Situation* KPI), meaning that a considerable heat stress condition exists for the flock. A HIS value higher than 81 triggers an emergency state (*Emergency Situation* KPI), indicating that extreme heat conditions exist.

The KPI values obtained for the previously presented case of real-world poultry production chain use breeding phase are shown in Table 1. As shown by the *Temperature Warnings* KPI, 18.31% of the time when the flock was within the farm, the temperature was moderately distant compared with the optimal. Furthermore, for the *Temperature Alarms* KPI, a value of 34.48 was obtained, which means that 34.48% of the time, the flock was exposed to potentially harmful situations where the temperature within the farm was very different from the desired one. This means that, overall, the flock has been exposed to undesirable temperatures more than half of the time it stayed in the farm. Regarding the heat stress determined by the HIS, 7.89% of the time the flock has been exposed to alert situations, 5.62% of the time to a more severe danger situation, and 68.30% to emergency situations. That is, the flock has been exposed to undesirable heat stress situations a combined 81.81% of the whole rearing period.

### 4.2. Loading Phase

The objective of the loading phase indicators is to identify the quality of the conditions in which the flock is loaded into transport trucks. Within the context of the IoF2020 project, the poultry catching and loading is performed manually, so, in this phase, the force with which the operators carry out these tasks is controlled. Force is directly proportional to acceleration (by Newton’s second law), therefore, operators arm sway acceleration is captured with electronic wristbands. These wristbands contain a sensor that measures acceleration on three axes, that is, they measure three components of acceleration: $a_{x},a_{y},a_{z}$. The acceleration is measured in g-force, or the gravitational force, which is a measurement of the type of force per unit mass that causes a perception of weight. A g-force of 1 g is equivalent to the conventional value of gravitational acceleration on Earth, about 9.8 m/s^2^. The wristband sensor has a range of forces that can measure, being −8.00 g the lowest measurable value, and 7.98 g the highest. Furthermore, the sensor has a sample rate of 1000 Hz, that is, it logs data with a frequency of 1 ms.

In order to determine the overall quality of a loading phase, 3 KPIs are defined. The first KPI called *Saturation Rate*, is based on the aggregation of the acceleration data on every second, and determines the average of the amount of acceleration data that exceeds the wristband sensor range of forces per minute. In other words, the average of the amount of saturated status. The remaining two KPIs are based on the acceleration module *a*, which is measured in *g* and is calculated as follows:
(1)$$a=\sqrt{a_{x}^{2}+a_{y}^{2}+a_{z}^{2}}$$


Namely, these two KPIs are the *Mean Accumulation* and the *Standard Deviation* of acceleration. They are both are based on the amount of acceleration every minute, calculated from the maximum value of acceleration module every second.

The KPI values obtained for the previously presented real-world poultry production chain use case’s transport phase are shown in Table 2. The 0.98 value for the *Saturation Rate* means that the wristband sensor gets saturated almost once per minute.

### 4.3. Transport Phase

In order to determine the overall quality of a transport phase, five KPIs are defined, which can be classified in two groups according to the environmental aspect that they specify. The first set of KPIs determines the flock’s thermal comfort and it is composed of 4 KPIs: *High Temperature*, *Low Temperature*, *High Relative Humidity* and *Low Relative Humidity*. These KPIs determine the percentage of the transport time under which the flock is exposed to temperatures over 31 °C (*High Temperature*) or below 18 °C (*Low Temperature*) and to relative humidity of over 80% (*High Relative Humidity*) or below 60% (*Low Relative Humidity*), which are considered to be stressful and harmful situations for the animals. The fifth KPI, the *Abrupt Movements* KPI, determines the quality of the transport in terms of the driver’s abruptness when driving. This KPI derives from the acceleration measurements of the IoF transport sensor deployed in the transport truck and calculates the percentage of time in which flocks suffer from the drivers sudden speed changes. In order to calculate this, a peak detection algorithm called The Smoothed Z-score Peak Detection Algorithm [57] is defined, based on the theoretical normal distribution of the acceleration. If a new acceleration measurement is a given x number of standard deviations away from a given moving mean, the algorithm generates a signal for this value.

This algorithm requires three parameters to be configured: lag, threshold and influence. The lag parameter is the size of the moving window and determines, on the one hand, the amount of data to be smoothed, and on the other hand, how adaptive the algorithm is to the changes in the long-term average of the data. The threshold parameter is the z-score at which algorithm generates signals, that is, the number of standard deviations from the moving mean above which the algorithm will classify a new data point as being a signal. Finally, the influence parameter indicates the effect of new signals on the mean and standard deviation and it has a value between 0 and 1. That is, it determines the effect of signals on the detection threshold of the algorithm. If value is 0, the signals have no influence on the detection threshold, therefore, future signals will be detected based on a detection threshold that is calculated with a mean and standard deviation that are not influenced by previous signals.

Let *a* be the acceleration module vector of length *t*, which is calculated with Equation (Equation 1), where $a_{x}$, $a_{y}$ and $a_{z}$ are the acceleration components measured in *g* along the X,Y,Z axes.

lOnce the acceleration module is calculated, the peak detection algorithm works as follows shown in Algorithm 1:
**Algorithm 1:** The Smoothed Z-score Peak Detection Algorithm.
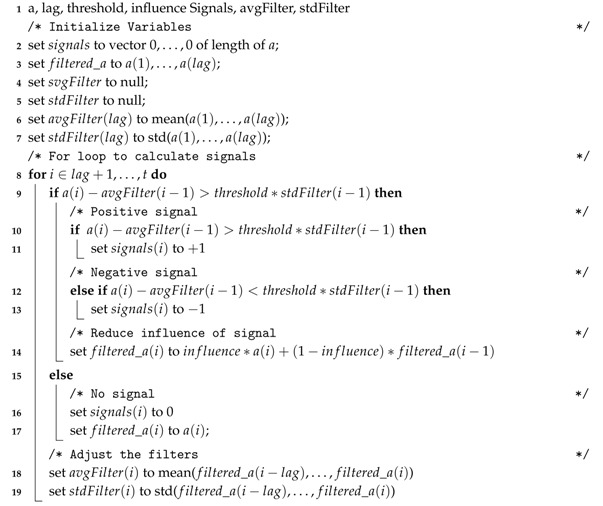


The algorithm returns three vectors called *signals*, *avgFilter* and *stdFilter*. The first one takes the values −1, 0 or +1 depending on whether the acceleration is considered stable (value 0) or not (values −1 or +1). If a new acceleration point exceeds the threshold of standard deviations away from the moving mean, it will assign +1 when it is above the upper limit, and −1 when it is below the lower limit. Otherwise, the algorithm will assign the value 0 to the signal variable. The other two correspond to the moving average (*avgFilter*) and standard deviation (*stdFilter*) of the previous data window, and they are calculated each time a new acceleration point is analyzed by the algorithm.

With views to ease the understanding to the Smoothed Z-score Peak Detection algorithm, an illustrative example will be provided next. Given lag=25, threshold=5 and influence=0 as parameters of the Smoothed Z-score Peak Detection Algorithm, and *a* the acceleration module calculated using the $a_{x}$, $a_{y}$ and $a_{z}$ data captured for 60 s, a simulation of that algorithm is computed. Acceleration values are shown in black in Figure 5. The blue discontinued line is obtained from avgFilter and two red discontinued lines are the limits of the confidence interval that is built using the algorithm. Finally, the signal results are represented in Figure 6 using a simple line graph to illustrate the abruptness score of the driver when transporting the animals.

Table 3 shows KPI values for the case of the previously presented use transport phase. It can be stated that temperature has been in comfort ranges as it has never reached values which are too high or too low. As for the relative humidity values measured, 17% of the transport time they have surpassed the upper comfort limit (i.e., 80%), thus exposing the flock to excessive relative humidity. Finally, the truck that transports the flock from the farm to the slaughterhouse has undergone abrupt changes of acceleration during 9.32% of the whole journey time.

### 4.4. Slaughterhouse Phase

In total, a set of 19 KPIs are defined to determine the slaughterhouse phase quality. This set includes KPIs to evaluate the physical conditions of the flock on its arrival (e.g., broken wings, bruises and broken bones) and the processing and packaging tasks performed (e.g., percentage of bad eviscerated and bad showered poultry). The last KPI, *Meat Quality*, takes only two categorical values depending on the quality of the product: *A* if the majority of the final meat is of a high-level, and *B* otherwise. These KPIs are manually evaluated by slaughterhouse operators, and stored in an Excel file that follows a predefined template.

Regarding the KPI values obtained for the slaughterhouse phase of the use case, they are shown in Table 4. These results refer to a sample of 200 chicken from a lot of 5040.

## 5. PUMA: A Decision Support System for Poultry Chain Managers

The PCM platform stores both the data collected through the poultry production chain and the generated KPIs in order to enable its further exploitation by different services and tools. One of those tools is PUMA (PoUltry Management Advisor), an Artificial Intelligent system inspired by the inherent requirements of the improvement of the whole poultry production chain. Namely, it is a Decision Support System for poultry chain managers developed with Shiny https://shiny.rstudio.com/, an R package that supports the building of interactive web apps straight from R. A screenshot of PUMA is shown in Figure 7.

PUMA is based on machine learning algorithms to extract knowledge from the KPIs generated through the different poultry production chain phases (explained in Section 4). It leverages a Decision Tree type of algorithm, namely Classification and Regression Tree (CART), which is a supervised learning method that uses a tree structure in order to go from features of an item (represented as branches) to conclusions about the item’s final value (represented as leaves). The CART algorithm is implemented using R’s Recursive Partitioning And Regression Tree (RPART) package https://www.rdocumentation.org/packages/rpart/versions/4.1-15. Decision trees can be thought as a disjunction of conjunctions that result in IF-THEN-ELSE type of rules. These rules can also be represented in a more formal way by means of Disjunctive Normal Forms (DNF). Furthermore, the use of a Decision Tree favors the transparency and the explainability in decision-making, thus aligning with the European Commission’s Ethics guidelines for trustworthy Artificial Intelligence systems https://ec.europa.eu/digital-single-market/en/news/ethics-guidelines-trustworthy-ai.

With views to making PUMA flexible and applicable to different use cases, it is developed so that it allows the manual selection of the variable to be predicted and the set of explanatory variables to be used for creating the decision tree. Furthermore, data used for the creation of the decision tree can be retrieved from a database or a CSV file. For example, a PUMA user may select the “number of deaths" variable as the feature to be predicted, and the transport phase KPIs to see how the transport conditions may influence the number of deaths of poultry in transport. For demonstration purposes, let us consider the following scenario: a chain manager wants to receive recommendations for the different poultry production chain phases in order to maximize the final meat quality. To do so, the Meat Quality variable will be selected as the feature to be predicted, and the Breeding, Loading and Transport KPIs as the explanatory variables.

PUMA leverages rules generated from decision trees derived from historical KPIs. As more poultry production chains are observed and the corresponding KPIs are calculated, decision trees and the derived rules are updated. The more the poultry production chains that are observed, the bigger the reliability of the generated rules. These rules capture the real behavior of the breeding carried out on that farm, their respective loads of the animals in the trucks, transport to the slaughterhouses, and their physical status on arrival as well as the final meat quality.

For the purpose of demonstrating PUMA, a set of poultry production chains were simulated and the corresponding set of KPIs were generated. The decision tree shown in Figure 8 is the result of this simulation.

Furthermore, the following rules expressed in DNF can be derived by searching a path through the created decision tree:
Rule (1): ES>65%∧AM>40%∧HH>20%→MQ=BRule (2): ES>65%∧AM>40%∧HH≤20%→MQ=ARule (3): ES>65%∧AM≤40%∧HT>50%→MQ=BRule (4): ES>65%∧AM≤40%∧HT≤50%→MQ=ARule (5): ES≤65%∧LT>35%→MQ=BRule (6): ES≤65%∧LT≤35%→MQ=A
where ES: Emergency Situation, AM: Abrupt Movements, HH: High Relative Humidity, HT: High Temperature, LT: Low Temperature and MQ: Meat Quality.

In order to showcase the use of this decision tree, the KPIs generated from a real-world poultry production chain use case (described in Section 4) will be used. In this use case, the value of ES is 60.30%, the preconditions of rules (5) and (6) cannot be satisfied and therefore they are discarded. The objective of the chain manager using PUMA is to get the A value for variable MQ, so rules (2) and (4) are those followed. Depending on the quality of the transport trucks and the routes followed by them, experts will decide between giving some recommendations or others. There are two ways to get meat to reach quality A, either AM>40 and HH≤20 (2) or AM≤40 and HT≤50 (4).

Let us consider that an optimal form to control the temperature is available in trucks but there is no adequate relative humidity control, so following the rule (4) is the best option to get high meat quality and the decision is made to take the truck along a smoother road, such as a highway, so that the AM KPI does not exceed 40%. The simulation results show that AM = 9.32% and HT = 0%, so this lot will be cataloged with the meat quality of type A.

## 6. Conclusions

The population of the world is growing at exponential rates and one of the main challenges consists of finding a way to feed all these people. One of the potential answers to this challenge considers meat production improvement, although it cannot be done at any cost. Maintaining the health and welfare status of animals at optimal levels has traditionally been one of the main concern of farmers, and, more recently, consumers. One of the most relevant types of meat is poultry, which is expected to grow at a worldwide level, reaching 181 million tonnes in 2050.

Although there are solutions that aim at monitoring different poultry production chain phases, there is no existing solution that collects data throughout the whole chain. In this article, the Poultry Chain Management (PCM) platform is presented. It collects data across the different phases of the poultry production chain in a centralized and secure way. The collection of this data establishes the base for the definition of a set of KPIs that determine the quality of each phase. Specifically, a total of 32 indicators are defined: 5 in the breeding phase, 3 in the loading phase, 5 in the transport phase, and 19 in the slaughterhouse phase. These KPIs contribute to determining the quality of each phase and the poultry production chain as a whole.

Furthermore, the exploitation of these KPIs paves the way towards the identification of critical issues causing process inefficiencies. As a matter of fact, this exploitation is demonstrated with services which exist already such as PUMA, which supports the decision-making towards the holistic improvement of the poultry production chain.

Last but not least, the PCM platform is based on open-source components and open standards, with views to make it reusable for other livestock production chains with minimum modifications that can be methodically approached and are expected to be of bounded complexity.

### Future Work

The contribution presented in this article tries to enhance the poultry production chain phases as well as the chain as a whole. However, it also opens up new paths for research and improvements of the existing solution.

On the one hand, the data collection of the poultry production chain in some phases could be automatized and better integrated with the PCM platform. Namely, the data generated in the slaughterhouse phase should ideally be collected in a seamless way. Instead of using an Excel file, an interface with a simple form could be developed, so that slaughterhouse operators could fill in the KPI information and send it to the PCM platform. This would contribute to simplifying the storage system design by having a unique centralized repository, instead of having a database and a file storage system.

On the other hand, the set of defined KPIs could be extended with additional ones that cover other relevant aspects affecting poultry welfare. For example, IoF environmental sensors installed in farms measure CO_2_ and ammonia levels (which are demonstrated to affect poultry welfare [58,59,60]), therefore, they could be exploited. Furthermore, platform design of the PCM enables the addition of new KPIs with minimal development requirements.

## Figures and Tables

**Figure 1 sensors-20-01549-f001:**
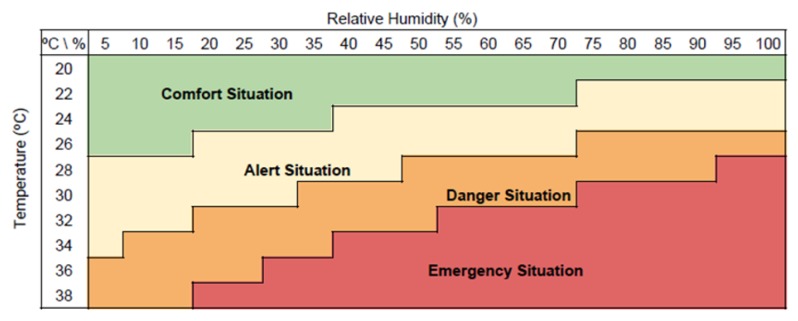
HIS (Heat Stress Index) for poultry [16].

**Figure 2 sensors-20-01549-f002:**
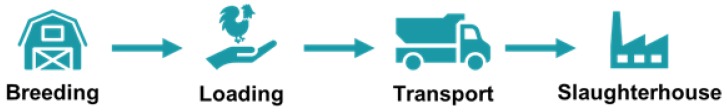
The four phases involved in a poultry production chain.

**Figure 3 sensors-20-01549-f003:**
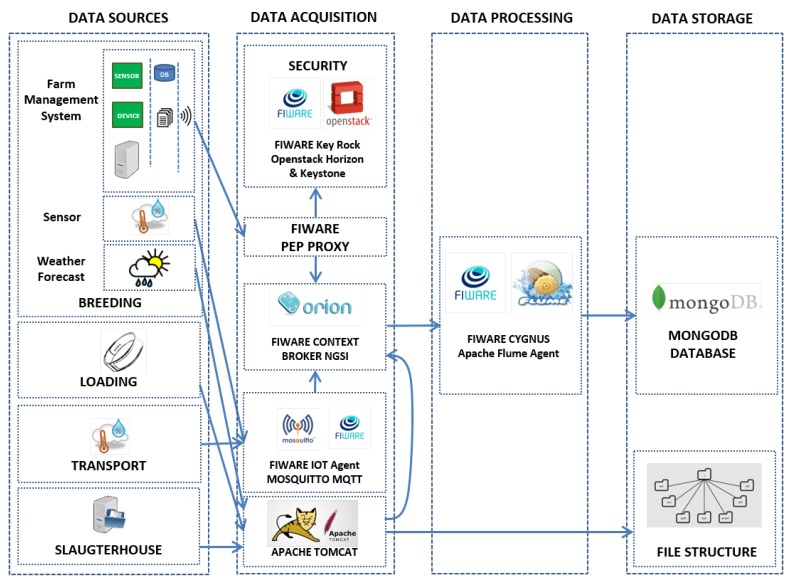
The PCM platform data flow.

**Figure 4 sensors-20-01549-f004:**
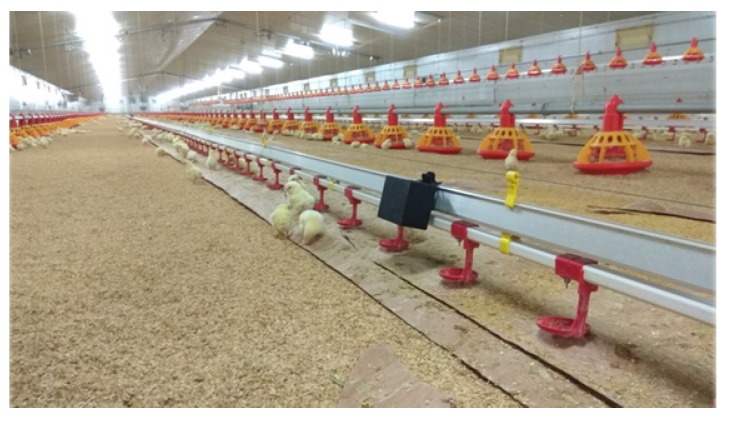
Real-world poultry farm used for demonstration purposes.

**Figure 5 sensors-20-01549-f005:**
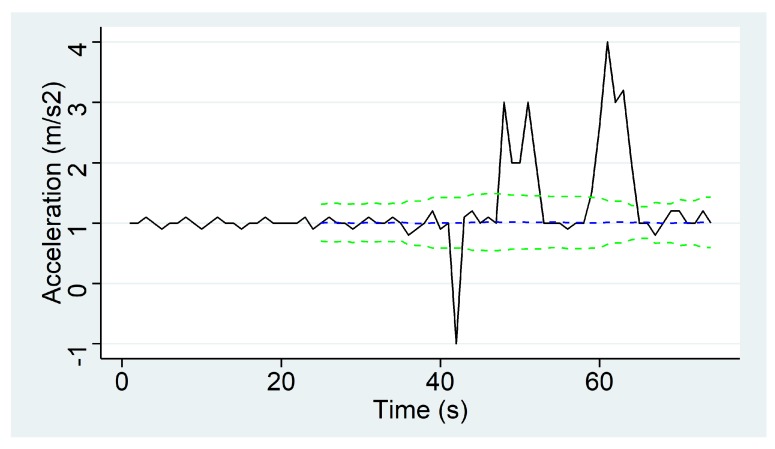
Results of simulation of the Z-Peak Algorithm.

**Figure 6 sensors-20-01549-f006:**
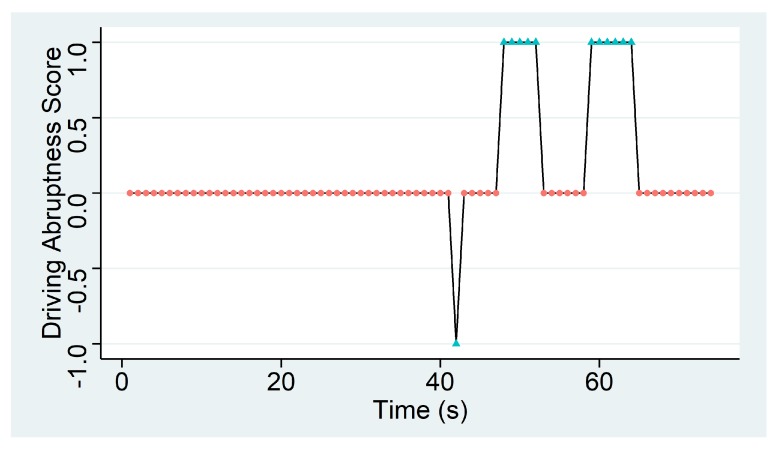
Representation of a driving abruptness score.

**Figure 7 sensors-20-01549-f007:**
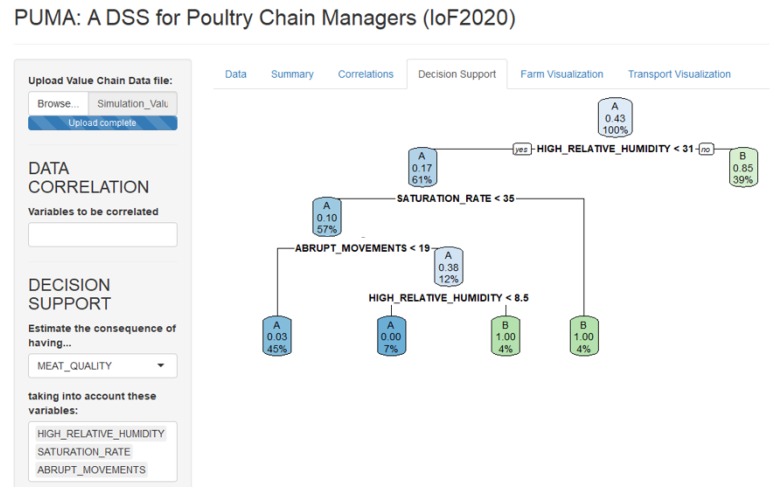
Screenshot of the PUMA tool.

**Figure 8 sensors-20-01549-f008:**
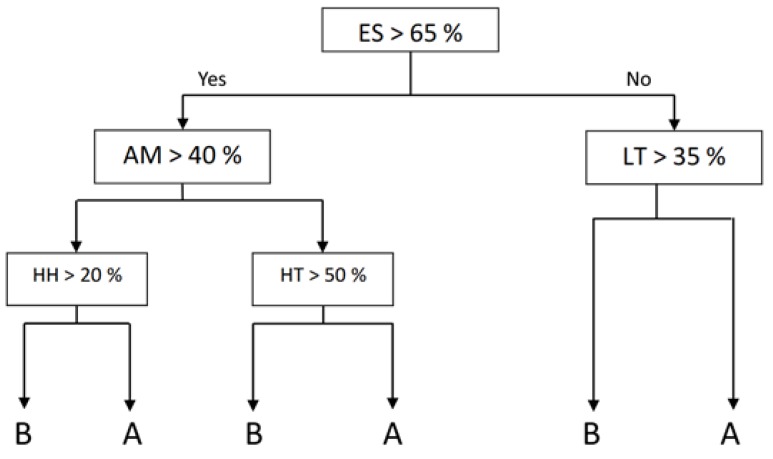
Decision Tree obtained training the algorithm with a simulated KPI.

**Table 1 sensors-20-01549-t001:** KPI results obtained in poultry breeding phase.

KPI	Value
Temperature Warning	18.31 %
Temperature Alarm	34.48 %
Alert Situation	7.89%
Danger Situation	5.62 %
Emergency Situation	68.30 %

**Table 2 sensors-20-01549-t002:** KPI results obtained in loading phase.

KPI	Value
Saturation Rate	0.98
Mean Accumulation	121.67
Standard Deviation	48.99

**Table 3 sensors-20-01549-t003:** KPI results obtained in transport phase.

KPI	Value
Low Temperature	0%
High Temperature	0%
Low Relative Humidity	0%
High Relative Humidity	17%
Abrupt Movements	9.32%

**Table 4 sensors-20-01549-t004:** KPI results obtained in slaughterhouse phase.

KPI	Value
Weight Range	3.42 Kg
Farm Weight	3.42 Kg
Total Hematoms	23
Broken Wing	21
Hematoma Wings	11
Hematoma Armpit	0
Breast	12
Broken Bones	9
Overscalded	0
Bad Extraction Viscera	1
Bad Plucked	0
Bad Wash	0
Scab	0
Crops	0
Knuckles	0
Dead in transport	3
Confiscated	8
Numbers of chickens	5040
Meat Quality	A

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
