# Peer review of "An IoT Platform towards the Enhancement of Poultry Production Chains"

_sensors, 2020, doi:10.3390/s20061549_

Round 1

Reviewer 1 Report

This is an interesting applied research study. Although I am not in the Food research domain, the work described in this paper is easy to follow. The only major concern of mine is Figure 7 and its related texts in Page 14. I am totally lost here. Is “smoothness” an indicator? If not, do you use smoothness to reflect Abrupt Movements? Then what is their relationship? What does Figure 7 tell in this case? What do the vertical axis and its unit mean?

The minor issues are mostly related to the reporting quality.

  1. The title of Section 3 is misleading. Here you don’t explain the poultry production chain, but the architecture/workflow of your developed poultry chain management platform.
  2. Section 3.1 is confusing. According to Figure 3, you are trying to describe the architecture of the PCM platform by specifying its layers. But the current description reads more like a workflow with its data processing stages.
  3. Line 345 in Page 10: “One of those generates…” One of those what?
  4. The caption of Figure 5 doesn’t make any sense. “X, Y and Z axes” can be anything.
  5. The texts in Figure 6, 7, and 8 are barely readable without zooming in.
  6. Line 544 in Page 17, duplicate words in “… reaching reaching 181…”

Author Response

First of all, we would like to thank the two reviewers for their time revising our article. Their comments and suggestions are much appreciated by authors.

Comments and Suggestions for Authors

This is an interesting applied research study. Although I am not in the Food research domain, the work described in this paper is easy to follow.

Thank you. We are glad to know this.

The only major concern of mine is Figure 7 and its related texts in Page 14. I am totally lost here. Is “smoothness” an indicator?  If not, do you use smoothness to reflect Abrupt Movements?  Then what is their relationship? What does Figure 7 tell in this case? What do the vertical axis and its unit mean?

Following your comment, we tried to rewrite this part in order to ease the reader’s understanding. The indicator we defined is the “Abrupt Movements”, and in order to avoid misunderstandings, we decided to remove the “smoothness” word and just stick with “abruptness” word. We also defined Figure 7’s vertical axis as “driving abruptness score”, where 0 is a non-abrupt (smooth) driving, and -1 or +1 mean abrupt driving.

The minor issues are mostly related to the reporting quality.

(1) The title of Section 3 is misleading. Here you don’t explain the poultry production chain, but the architecture/workflow of your developed poultry chain management platform.

The initial paragraphs of the Section 3 try to explain the whole poultry production chain, subsections 3.1 the PCM platform, and subsection 3.2 the data collected from each phase. However, your comment is reasonable. The reader may be confused expecting solely information related to the poultry production chain. Following your comment, we modified the title of section 3 to “The Poultry Chain Management Platform”.

(2) Section 3.1 is confusing. According to Figure 3, you are trying to describe the architecture of the PCM platform by specifying its layers. But the current description reads more like a workflow with its data processing stages.

Following your comment, we modified subsection 3.1 to “The PCM Platform Workflow”, as well as Figure 3’s caption to “The PCM platform data flow”.

(3) Line 345 in Page 10: “One of those generates…” One of those what?

Following your comment, we specified the sentence: “One of those services”

(4) The caption of Figure 5 doesn’t make any sense. “X, Y and Z axes” can be anything.

We thought that Figure 5 didn’t contribute much to the understanding of the section, so we decided to remove it.

(5) The texts in Figure 6, 7, and 8 are barely readable without zooming in.

Following your comment, we tried to make them more readable by making them bigger.

(6) Line 544 in Page 17, duplicate words in “… reaching reaching 181…”

We were not aware of this type. Following your comment, we removed the duplicated word.

Reviewer 2 Report

The paper presents a platform, named Poultry Chain Management (PCM). It aims at collecting data across the different phases of the poultry production chain, in order to perform quality checks and to identify critical issues. To this end, also a support to the decision-making task is provided by integrating the tool PUMA (PoUltry Management Advisor).

In general, the paper is well-written and well-organized.

Some concerns are the following:

  • At the end of Section 2, it would be useful to add a table where the discussed works are compared, with respect to the solution proposed in the paper.
  • Some aspects of the platform still deserve more attention, such as security.

Author Response

First of all, we would like to thank the two reviewers for their time revising our article. Their comments and suggestions are much appreciated by authors.

Comments and Suggestions for Authors

The paper presents a platform, named Poultry Chain Management (PCM). It aims at collecting data across the different phases of the poultry production chain, in order to perform quality checks and to identify critical issues. To this end, also a support to the decision-making task is provided by integrating the tool PUMA (PoUltry Management Advisor).

In general, the paper is well-written and well-organized.

Thank you. We are glad to know this.

Some concerns are the following:

  • At the end of Section 2, it would be useful to add a table where the discussed works are compared, with respect to the solution proposed in the paper.

We think that this kind of tables are definitely helpful for somehow organizing the presented related work and see which is the advance proposed. However, in this article, we find it difficult to create such tables, since in most of the poultry chain phases, we are not presenting an advancement in terms of how they are measured. Actually, except for a few KPIs, the rest are based on existing ones. Therefore, we decided that the best option was to leave this section as it is, in order to avoid possible misunderstandings from the readers.

  • Some aspects of the platform still deserve more attention, such as security.

Following your comment, we rewrote the security part of the PCM platform, trying to emphasize its importance.